# Microstructure and Mechanical Properties of High-Specific-Strength (TiVCrZr)_100*−x*_W*_x_* (*x* = 5, 10, 15 and 20) Refractory High-Entropy Alloys

**DOI:** 10.3390/e25010100

**Published:** 2023-01-03

**Authors:** Haitao Wang, Kuang Xu, Juchen Zhang, Junsheng Zhang

**Affiliations:** 1School of Mechanical and Electrical Engineering, Shenzhen Polytechnic, Shenzhen 518055, China; 2School of Mechanical Engineering, Hefei University of Technology, Hefei 230009, China

**Keywords:** refractory high-entropy alloy, microstructure, specific strength, phase

## Abstract

With the increasing demand for high-specific-strength materials for high-temperature applications, particularly in the aerospace field, novel (TiVCrZr)_100*−x*_W*_x_* (*x* = 5, 10, 15 and 20) refractory high-entropy alloys (RHEAs) were developed. The phase formation, microstructure, and mechanical properties were studied. The (TiVCrZr)_100*−x*_W*_x_* RHEAs exhibit a relatively high specific strength and low density compared with the W-containing RHEAs and most of the W-free RHEAs. In (TiVCrZr)_100*−x*_W*_x_* RHEAs, Laves, BCC and Ti-rich phases are formed, where the Laves phase is the major phase, and the volume fraction of the BCC phase increases with increasing W content. (TiVCrZr)_100*−x*_W*_x_* RHEAs exhibit dendrite structures, where W is enriched in the dendrite region, and increasing W-rich precipitations corresponding to the BCC phase are observed. The improvement of the strength and hardness of RHEAs is mainly attributed to the evolution of the microstructure and corresponding strengthening effect of W. The empirical parameters and calculated phase diagram were investigated, which further explain and verify the formation and variation of phases. The present findings give more insights into the formation of multi phases in (TiVCrZr)_100−*x*_W*_x_* RHEAs, and explore their application potential in the aerospace industry and nuclear reactors due to their high specific strength and low-activation constituent elements.

## 1. Introduction

High-entropy alloys (HEAs), firstly proposed in 2004, have attracted extensive attention due to their outstanding mechanical properties, corrosion resistance, irradiation resistance and thermal stability [1,2,3,4,5]. HEAs are generally composed of at least five kinds of elements with an atomic percentage between 5% and 35% in equal mole or near equal mole ratios [6,7]. Differing from conventional alloys, HEAs tend to form simple solid solution structures, such as face-centered cubic (FCC), body-centered cubic (BCC) and close-packed hexagonal (HCP), rather than intermetallic compounds or amorphous phases due to their higher mix entropy [8,9,10,11]. However, the existing HEAs also contain several intermetallic compounds, which can strengthen the mechanical properties [12,13]. With the urgent demand for structural materials for application in high-temperature conditions, refractory high-entropy alloys (RHEAs), generally composed of refractory elements, were proposed in 2010 [14]. Senkov et al. [15] studied the microstructure and mechanical properties of WNbMoTa and WNbMoTaV RHEAs; the results showed that these two RHEAs have a typical BCC structure, and exhibit a higher strength and hardness especially at elevated temperature. RHEAs are therefore considered as promising next-generation high-temperature materials, which show great application potential in the nuclear fusion and aerospace fields [16,17].

High-specific-strength materials play a critical role in engineering applications, especially for the aerospace and automobile industries, as they can reduce the weight of structural parts and achieve energy conservation [18,19]. The RHEAs reported in previous studies usually have a higher density and lower specific strength [14,15,16], which restrict their engineering applications. Thus, the development of RHEAs with high specific strength has become a critical issue. The unique elemental composition of RHEAs provides extensive freedom when designing alloys with desired performances and tuning properties [20,21,22]. Senkov et al. [23,24] selected refractory elements with low density including Ti, V, Cr, Zr and Nb, and developed the CrNbTiVZr alloy system. The density of the CrNbTiVZr alloy system ranges from 6.34 g/cm^3^ to 6.67 g/cm^3^, and the CrNbTiVZr RHEA has a maximum specific strength approaching 200 MPa·cm^3^·g^−1^. Although the CrNbTiVZr alloy system has a low density, the specific strength still needs to be further improved. Benefiting from the abundant elemental composition of RHEAs, adjusting the type and content of elements has been widely used in the development of high-specific-strength RHEAs. Stepanov et al. [25] studied the mechanical properties of Al*_x_*NbTiVZr HEAs. With the increase of Al content, the density of the alloy system is further reduced, where Al_1.5_NbTiVZr HEA has the minimum density of 5.55 g/cm^3^. However, Al addition results in a decrease of melting point.

In this work, in order to achieve higher specific strength in RHEAs for high-temperature applications, Ti, V, Cr and Zr refractory elements with low density were selected for the design of the RHEAs. On the other hand, W has a significant strengthening effect on the mechanical properties of RHEAs, especially for the achievement of a high melting point [26,27]. For instance, Niu et al. [27] investigated the microstructure and mechanical properties of CoCrFeNiW*_x_* HEAs. With the addition of W, the compositional phases change from FCC phase to FCC and μ-(FeCoCr)_7_W_6_ phases, and the microstructure transforms from coarse columnar crystal to dendritic and eutectic laminar structure, improving the hardness and strength. Therefore, W was selected as the element variable here. Based on the above, (TiVCrZr)_100−*x*_W*_x_* RHEAs were developed, and the microstructure and mechanical properties were also investigated. The results demonstrate that (TiVCrZr)_100−*x*_W*_x_* RHEAs exhibit a relatively high specific strength, where the maximum specific strength of the (TiVCrZr)_90_W_10_ RHEA can reach 256.65 MPa·cm^3^·g^−1^. The hardness, strength and melting point of RHEAs are significantly enhanced with the addition of W. The corresponding strengthening mechanisms are also discussed.

## 2. Materials and Methods

(TiVCrZr)_100−*x*_W*_x_* (*x* = 5, 10, 15 and 20, denoted as W5, W10, W15 and W20 respectively) RHEAs were prepared by vacuum arc melting of the raw metals with a purity higher than 99.95% under a Ti-getter argon atmosphere. Each alloy ingot was remelted 12 times to ensure its homogeneity. The phase formation of RHEAs was characterized by a PANalytical X’Pert PRO MPD X-ray Diffractometer. The microstructures of RHEAs were inspected by a field emission scanning electron microscope (FESEM, Hitachi SU8020, Japan), where samples were polished by abrasive papers (Matador, Germany) with different grits from 400 to 5000. The attached energy dispersive spectrometer (EDS) was employed to inspect the elemental content and distribution of RHEAs. The Vickers hardness values of the RHEAs were measured by a Vickers hardness tester (WeiYee, HV-1000SA, China) under 1000 gf loads lasting 10 s, and the hardness values of 10 different positions for each sample were collected to obtain the average value. The samples were cut from a button-shaped ingot by a wire electrical discharge machine. The sample surfaces were polished by abrasive papers (Matador, Germany) with different grits from 400 to 2000. The typical W10 RHEA was annealed at 1000 °C for 10 h under argon atmosphere, where the heating rate was 10 °C/min. Compression tests of RHEAs were conducted by an electronic universal material testing machine (WanCe, DNS100, China) at an initial strain rate of 1 × 10^−3^ s^−1^ at room temperature, where the samples’ dimensions were 3 mm × 3 mm × 6 mm [28]. The fracture morphology of samples was also observed by the FESEM.

## 3. Results

### 3.1. Phase Characterization of (TiVCrZr)_100−x_W_x_ RHEAs

According to the X-ray diffraction (XRD) patterns (Figure 1), (TiVCrZr)_100−*x*_W*_x_* RHEAs have three different phases including the Laves (cF24-Cu_2_Mg, SG No. 194) [29], BCC and Ti-rich phases. Particularly, the W5 RHEA only has the Laves and Ti-rich phases. It is emphasized that the main peaks of the BCC and Laves phase are close to coincidence. The Laves phase is the major phase based on the diffraction intensity of the peak. It is worth noting that there were two peaks in the main peak of the Laves phase (Figure 1b), and with the increase of W content, the intensity of the left peak gradually increased, while the intensity of the right peak gradually decreased. For W15 and W20 RHEAs, the right peak was almost invisible. It is speculated that there are two Laves phases with slightly different compositions, and a Laves phase gradually disappears while another Laves phase gradually increases with increasing W content. A detailed discussion is presented in a later section. Moreover, the BCC increased based on the intensity of the corresponding diffraction peak. For the W20 RHEA, the main peak shifted obviously right, which may be influenced by the formation of more BCC phase.

### 3.2. Microstructure of (TiVCrZr)_100−x_W_x_ RHEAs

The microstructure of (TiVCrZr)_100−*x*_W*_x_* RHEAs is shown in Figure 2. (TiVCrZr)_100−*x*_W*_x_* RHEAs exhibit a typical dendrite structure, which widely exists in HEAs due to the element segregation [30,31]. For the W5 RHEA, the microstructure comprised three distinct constituents, having bright, gray and dark regions. As shown in Figure 2a, the gray region was mainly distributed around the bright region, for example, the marked region between the blue and yellow dotted line. The fine gray region was dispersed in the interdendrite region. With the increase of W, the gray region around the bright region (W5) turned to a bright region, corresponding to the disappearance of the gray region around the bright region (W10, W15 and W20). However, a small amount of fine gray region dispersed in interdendrite regions was still observed. It is worth noting that the white region appeared inside the dendrite in W10, W15 and W20 RHEAs, and the proportion of the white region increased with the addition of W, corresponding to the appearance and enhancement of the BCC phase diffraction peak. Based on the XRD patterns and SEM images, the bright region, corresponding to the Laves phase, occupied most area of the microstructure, and the white region corresponds to the BCC phase, while the dark region corresponds to the Ti-rich phase. The gray region will be analyzed in detail in the subsequent section.

EDS analysis of different regions was conducted to study the elemental distribution and further verify the types of each phase. As shown in Figure 3, V was distributed uniformly in the whole region, while other elements occur to different degrees of segregation due to their different melting points [32]. W was enriched in the dendrite region, particularly the white region, which is similar to previous studies [26,27,33]. This may result from its high melting point and limited solubility. The difference of W concentration between the bright and gray region drove the diffusion of W from the bright to gray region, resulting in the disappearance of the gray region around the bright region. With the increase of W content, W precipitated when the concentration of W in the dendrite region reached a certain limit, corresponding to the white region in Figure 2. In the solidification process of alloys, high melting point elements like W, Ta, Mo and Nb favor nucleate preferentially, resulting in the enrichment of elements in distinct regions [16,28,32]. However, Ti was mainly enriched in the interdendrite region. This may be attributed to the first nucleation of high melting point elements, leading to the latter solidified liquid with more Ti content, and further formation of a Ti-rich interdendrite region. Moreover, the interdendrite region was depleted of Cr, while the white region was depleted of Zr.

In order to obtain the specific element content of distinct regions, EDS point tests were carried out, and three to five points of each region were inspected to obtain their average values. The results are shown in Table 1. As mentioned above, the phase corresponding to the gray region still needed to be analyzed. According to the elemental content of the gray region, it can be found that the content of V, Cr, Zr satisfied the formula (relation) of Zr(V, Cr)_2.04_, which is almost consistent with the form of pseudo-binary system Zr(V_1−*x*_Cr*_x_*)_2_ Laves intermetallic compound [34,35]. In general, the Laves phase, as a topologically close-packed (TCP) phase, is an intermetallic compound with a chemical formula of the main AB_2_ type, where the ratio of the atomic radius of elements A and B ranges from 1.09 to 1.55 [29]. For instance, ZrV_2_ and ZrCr_2_ Laves phases existing in the HEAs have been reported [36,37,38]. The appearance of the pseudo-binary system Zr(V_1−*x*_Cr*_x_*)_2_ Laves phase may be attributed to the mutual substitution (replacement) of Cr and V. As mentioned earlier, the bright region also corresponded to the Laves phase (cF24-Cu_2_Mg, SG No. 194). The EDS results of the bright region showed that the content of V, Cr, Zr and W satisfied the formula (relation) of Zr(V, Cr, W)_2.05_, a quaternary Laves, which may result from the replacement of V or Cr by W. Additionally, Senkov et al. studied the microstructure and phase formation of CrNbTiVZr RHEA, and the experimental results showed that the Laves phase is also not entirely the binary ZrCr_2_ Laves phase, but also contains other compositional elements. The actual composition of the Laves phase fulfills the formula (Cr, Nb, V)_2_(Ti, Zr), which indicates that part of Cr was substituted by V and Nb, and part of Zr was substituted with Ti. As a result, both the gray and bright regions belong to the Laves phase, and the difference between them was the W content. As shown in Figure 1b (B-spline curve fitting), there were two diffraction peaks in the main peak of the Laves phase, corresponding to two Laves phases with slightly different compositions. With the increase of W, the gray region disappeared due to the diffusion of W, and accordingly the right diffraction peak also disappeared, as shown in Figure 1b (B-spline curve fitting).

Moreover, the difference of atomic sizes of V, Cr and W was less than 15%, and V, Cr and W had similar BCC crystalline structures, which accorded with the replacement condition. Additionally, considering the enthalpy of mixing between pairs of atoms, the calculated results demonstrate that the enthalpy of mixing of Zr-W (−9 kJ/mol) ranked only second to that of Zr-Cr (−12 kJ/mol) in alloys, as shown in Figure 4. Large negative enthalpy of mixing usually represents strong attraction, i.e., combination tendency between atoms [39]. It is possible that the attraction of Zr to W leads to the substitution of Cr and V by W due to the relatively large negative enthalpy of mixing of Zr-W. However, the white precipitation was enriched with W while depleted of Zr, which may be attributed to the large difference in melting point.

Overall, the content of W in the dendrite region gradually increased, while the elemental content of interdendrite almost remained the same. The variation of microstructure in the dendrite region has a critical effect on the mechanical properties of (TiVCrZr)_100−*x*_W*_x_* RHEAs.

### 3.3. Mechanical Properties of (TiVCrZr)_100−x_W_x_ RHEAs

The compressive stress–strain curves of (TiVCrZr)_100−*x*_W*_x_* RHEAs at room temperature are shown in Figure 5. The corresponding fracture strength (*σ*_b_) and strain (*ε*_f_) are listed in Table 2. With the increase of W content, the fracture strength (*σ*_b_) and strain (*ε*_f_) were gradually enhanced except for the W20 RHEA. The strength of the alloy system increased from an initial 1574 MPa (W5) to a maximum of 1894 MPa (W15), while the strength of W20 (1754 MPa) decreased with further increase of W content. The high strength and low ductility of the alloy system are mainly attributed to the formation of the Laves phase, which exhibited a high strength and brittleness at room temperature. The fitting results (Figure 5c) of hardness and *x* value indicate that the hardness of the alloy system increased nearly linearly with the increase of *x* value, where the goodness of fit *R*^2^ = 0.958 was close to 1. This implies a highly linear relationship between the hardness and *x* value [40]. With the increase of W, the W content of the dendrite region gradually increased, particularly with increased W content of the Laves phase, resulting in the increase of hardness. For the W20 RHEA, although a large amount of W-rich phase led to a decrease of the Laves phase, the grain size was reduced and the high hardness of the W-rich phase still improved the hardness of the alloys. The addition of W can therefore significantly improve the strength and hardness of alloys over a certain range.

The above-mentioned mechanical properties results show that (TiVCrZr)_100−*x*_W*_x_* RHEAs exhibit high strength and brittleness due to the formation of the Laves phase. As shown in Figure 6, (TiVCrZr)_100−*x*_W*_x_* RHEAs exhibit a typical brittle fracture morphology, which is similar to the fracture morphology observed in CrNbTiVZr RHEA [24]. Some particles were observed on the fracture surface, particularly for W5 and W10 RHEAs, which may be attributed to the catastrophic fracture of the brittle Laves phase. It is worth noting that some spherical particles and a cluster of spherical particles were also observed. This fracture characteristic of alloys resulted from the instantaneous release of elastic energy, which led to the melting of the local region. According to the stress-strain curves of (TiVCrZr)_100−*x*_W*_x_* RHEAs, there were a large amount of accumulated elastic energy in specimens during the compressive process, and light emission appeared at the moment of fracture.

Moreover, after annealing at 1000 °C for 10 h, the W10 RHEA slightly demonstrated increased hardness and strength compared with the as-prepared specimens. As shown in Table 3, the hardness increased from 639 ± 23 HV to 647 ± 13 HV, while the strength increased from 1867 ± 52.5 MPa to 1930 ± 45 MPa. This indicates that the (TiVCrZr)_100−*x*_W*_x_* RHEAs may have better mechanical properties at high temperature. As shown in Figure 7a, the microstructure of the annealed specimens still exhibited four different contrasts including white, bright, gray and dark regions, which is similar to the microstructure of as-prepared specimens (Figure 7b). Thus, the (TiVCrZr)_100−*x*_W*_x_* RHEAs may possess higher thermostability and have potential for high temperature applications.

## 4. Discussion

### 4.1. Phase Structure

The experimental results indicate that (TiVCrZr)_100−*x*_W*_x_* RHEAs exhibit a complex phase formation rather than the simple solid solution as some HEAs exhibit. Empirical parameters and calculated phase diagram were employed for further understanding of the phase formation. Generally, HEAs tend to form solid solution phases due to the high entropy of mixing, while partial existing HEAs also tend to form Laves, σ, μ and χ phases of intermetallic compounds [41]. Particularly, the Laves phase is usually formed in Cr-containing and Zr-containing HEAs [23,42,43]. The phase type has a critical effect on the properties of HEAs, which are related to the thermodynamic, electronic parameters and atomic size of the alloy system [29,44]. The corresponding parameter values of compositional elements are listed in Table 4.

Empirical parameters including the entropy of mixing (Δ*S*_mix_), enthalpy of mixing (Δ*H*_mix_) and atomic size difference (*δ*) were proposed, which can be calculated by the following formulas [46]:(1)$$\Delta S_{\mathrm{mix}}=-R\overset{n}{\underset{i=1}{\sum }}c_{i}\ln c_{i}$$
(2)$$\Delta H_{\mathrm{mix}}=\overset{n}{\underset{i=1}{\sum }}4\Delta H_{\mathrm{ij}}^{\mathrm{mix}}c_{i}c_{j}$$
(3)$$\delta =\sqrt{\overset{n}{\underset{i=1}{\sum }}c_{i}{\left(1-r_{i}/r_{a}\right)}^{2}}$$
where *R* is the gas constant; *n* is the number of elements; *c*_i_ and *c*_j_ are the atomic percentage of the *i*th and *j*th elements; $\Delta H_{\mathrm{ij}}^{\mathrm{mix}}$ is the enthalpy of mixing between the *i*th and *j*th elements, which can be obtained by Ref. [47]; *r*_i_ is the atomic radius of the *i*th element; and $r_{a}=\overset{n}{\underset{i=1}{\sum }}c_{i}r_{i\;}$ is the average atomic radius. Then, Yang et al. [48] optimized the aforementioned rules and proposed a novel parameter *Ω*, which can be calculated by the following formulas:(4)$$\Omega =\frac{T_{m}\Delta S_{\mathrm{mix}}}{\left|\Delta H_{\mathrm{mix}}\right|}$$
(5)$$T_{m}=\overset{n}{\underset{i=1}{\sum }}c_{i}{\left(T_{m}\right)}_{i}$$
where *T*_m_ is the average melting point of the alloy and (*T*_m_)_i_ is the melting point of the *i*th element. The result shows that the formed phase types can be divided into four categories including solid solutions; intermetallics; bulk metallic glasses; and solid solutions and intermetallics in terms of the relationship of *Ω* and *δ*. The corresponding calculated values of (TiVCrZr)_100−*x*_W*_x_* RHEAs are distributed in the region of formation of solid solutions and intermetallics. However, these parameters cannot predict the specific type of phases. Guo et al. [49] studied the effect of valence electron concentration (*VEC*) on the phase type of HEAs, which can be calculated by the following formula:(6)$$VEC=\overset{n}{\underset{i=1}{\sum }}c_{i}{\left(VEC\right)}_{i}$$
where (*VEC*)_i_ is the valence electron concentration of the *i*th element, referring to Ref. [45]. The results indicate that HEAs tend to form a BCC phase or FCC phase with *VEC*< 6.87 or *VEC* ≥ 8 respectively. When 6.87 ≤ *VEC* < 8.0, BCC and FCC phase coexist in HEAs. Yurchenko et al. [50] studied the effect of Allen electronegativity difference (Δ*χ*_Allen_) and *δ* on the phase type of HEAs, which can be calculated by the following formula:(7)$$\Delta \chi_{\mathrm{Allen}}=\sqrt{\overset{n}{\underset{i=1}{\sum }}c_{i}{\left(1-\chi_{i}^{\mathrm{Allen}}/\chi_{a}\right)}^{2}}$$
where $\chi_{i}^{\mathrm{Allen}}$ is the Allen electronegativity of the *i*th element and $\chi_{a}=\overset{n}{\underset{i=1}{\sum }}c_{i}\chi_{i}^{\mathrm{Allen}}$ is the average Allen electronegativity. The results show that the Laves phase tends to form in HEAs when *δ* > 5.0% and Δ*χ*_Allen_ > 7.0% [50]. Therefore, the *VEC*, *δ* and Δ*χ*_Allen_ values shown in Table 5 demonstrate that (TiVCrZr)_100−*x*_W*_x_* RHEAs have the Laves and BCC phases, which is consistent with the experimental results.

Beyond empirical criteria, the calculated phase diagram is also widely applied in the analysis of the phase formation of HEAs. In this work, thermo-Calc (v. 2019a, Thermo-Calc Software AB, Stockholm, Sweden) in combination with the HEAs thermodynamic database (v. 2.1.1) were used to simulate the equilibrium solidification process. The calculated phase diagram also indicates that (TiVCrZr)_100−*x*_W*_x_* RHEAs exhibit a complex phase formation, including the Laves, BCC and HCP phases. Although the calculated phase diagram cannot entirely be consistent with the experimental results, some of the results still have reference value. As shown in Figure 8, (TiVCrZr)_100−*x*_W*_x_* RHEAs exhibit the ordered intermetallic compound phase in the as-cast state despite the high entropy of mixing. Nevertheless, (TiVCrZr)_100−*x*_W*_x_* RHEAs exhibit a mixture of single BCC solid solution and liquid phases at a higher temperature, which can be explained from the viewpoint of Gibbs free energy. Gibbs free energy has a critical effect on the formation of phases, which can be expressed by the following formula [12]:*G = H − TS*(8)
where *H* is the enthalpy; *T* is the thermodynamic temperature; and *S* is the entropy. According to Formula (8), the effect of entropy on Gibbs free energy becomes more significant under high temperature conditions, and can effectively reduce the Gibbs free energy of the system [51]. Lower Gibbs free energy can stabilize the solid solution. For (TiVCrZr)_100−*x*_W*_x_* RHEAs, the single BCC solid solution only exists in a narrow temperature range. However, the temperature range of a single BCC solid solution increases with increasing mixing entropy, resulting from the addition of W as shown in Table 5. This can be explained by the effect of the entropy increase on the reduction of Gibbs free energy. With the decrease of temperature, the Laves phase appears and compresses the single BCC solid solution range. The Laves phase becomes the major compositional phase until complete solidification. The equilibrium phase diagram shows that the fraction of the Laves phase decreases and the fraction of BCC phase increases with the increase of W content, which is consistent with the evolution of microstructure, i.e., the gradual precipitation of the white region corresponding to the BCC phase. It is worth noting that the Laves phase exists over a larger temperature range and the stable temperature range of the Laves phase increases with increasing W content. Additionally, the temperature at which the Laves phase starts to dissolve and the temperature at which the Laves phase completely dissolves also gradually increase. As mentioned earlier, W substitutes Cr and V to form a similar (Cr, V, W)_2_Zr-type Laves phase, resulting in an improved melting point of the Laves phase. The calculated phase diagram can predict the existence and variation trend of the Laves and BCC phases more clearly, while the HCP phase was not observed according to the XRD patterns. Nevertheless, the XRD results and microstructure show the existence of the Ti-rich phase. Indeed, Ti exhibits a HCP crystal structure at low temperature, and the equilibrium phase diagram simulates the formed phases at a slow cooling rate [32], which may result in the existence of a HCP phase as shown in Figure 8.

### 4.2. Specific Strength

Materials with a high specific strength have attracted extensive attention due to their huge advantages in engineering applications. (TiVCrZr)_100−*x*_W*_x_* RHEAs have high specific strength as shown in Table 6, exhibiting great application potential. In order to better reveal the behavior of high specific strength, the following contents will be discussed. On the one hand, (TiVCrZr)_100−*x*_W*_x_* RHEAs comprise four low-density refractory elements, which make the alloy system have a lower density. The density of (TiVCrZr)_100−*x*_W*_x_* RHEAs was calculated by the rule of mixture, which can be expressed by the following formula [16]:(9)$$\rho_{\mathrm{mix}}=\frac{\sum_{i=1}^{n}c_{i}A_{i}}{\sum_{i=1}^{n}\frac{c_{i}A_{i}}{\rho_{i}}}\;$$
where *c*_i_ is the atomic percentage of the *i*th element; *n* is the number of elements; *A*_i_ is the atomic weight of the *i*th element; and *ρ*_i_ is the density of the *i*th element. The theoretical density and actually measured density results are almost consistent with previous research [23,32], and thus the theoretical density was employed in this work. As shown in Formula (9), the low density of components in the alloy system can reduce the whole density of the alloys.

On the other hand, W addition significantly improves the strength of the alloy system while obtaining low density. With the increase of W content, the W content in the dendrite region increases, and the initial gray region (Zr(V_1−*x*_Cr*_x_*)_2_ Laves phase) around the bright region transforms into a bright region (Zr(V, Cr, W)_2_ Laves phase). It is known that HEAs have a cocktail effect, which implies that HEAs can sufficiently utilize the comprehensive properties of compositional elements. W possesses high strength and hardness [52,53], which has a positive effect on the mechanical properties of the alloy system. As a result, the bright region composed of the Zr(V, Cr, W)_2_ Laves phase has a higher strength and hardness than that of the gray region composed of the Zr(V_1−*x*_Cr*_x_*)_2_ Laves phase. The enhancement of the major phase (Laves phase), benefiting from W addition, improves the hardness and strength of the alloy system. For the W20 RHEA, a larger number of the W-rich BCC phase (brittle phase) precipitates in the dendrite region, leading to increasing brittleness at room temperature and reduced volume fraction of the Laves phase. These result in the decrease of the exhibited strength of the W20 RHEA. Although W addition also increases the density of the alloy system, the strengthening effect of W on the strength exceeds that of W on the density of the alloy system over a certain range. Therefore, with the increase of W, the specific strength of the alloy system first increases and then decreases, where W10 RHEAs can reach a maximum specific strength of 256.65 MPa·cm^3^·g^−1^.

To further understand the specific strength of (TiVCrZr)_100−*x*_W*_x_* RHEAs, some RHEAs were selected as comparative materials, as these RHEAs meet the following conditions. The involved RHEAs were prepared by vacuum arc melting. The selected RHEAs comprised W-containing RHEAs and W-free RHEAs, where W-free RHEAs consisted of nine refractory elements and contained any three of the four elements (Ti, V, Cr, Zr). As shown in Figure 9, (TiVCrZr)_100−*x*_W*_x_* RHEAs exhibited a higher specific strength and lower density, particularly for W10 RHEAs. Overall, the specific strength of (TiVCrZr)_100−*x*_W*_x_* RHEAs exceeded almost all W-containing RHEAs prepared by vacuum arc melting. The specific strength of the W10 RHEA was only inferior to the two W-free RHEAs among the selected RHEAs, i.e., Mo_0.5_VNbTiCr_1.5_ and Mo_0.5_VNbTi Cr_2.0_. As a result, (TiVCrZr)_100−*x*_W*_x_* RHEAs show great application potential in the aerospace industry due to their high specific strength. Additionally, the compositional elements of the alloys are low activation elements, and tungsten and its alloys possess great application prospects [40,54,55], implying that (TiVCrZr)_100−*x*_W*_x_* RHEAs may have excellent irradiation resistance. (TiVCrZr)_100−*x*_W*_x_* RHEAs not only have great application potential for turbine blades, but also outstanding application value in structural materials for nuclear reactors. Improving the plasticity of the (TiVCrZr)_100−*x*_W*_x_* RHEAs could expand their engineering applications. However, improving the plasticity is challenging due to the strength–ductility trade-off. Many ways have been proposed to further improve the plasticity by maintaining the strength [28,56,57]; for example, Sun et al. [56] tuned elemental composition to minimize the Gibbs free energy, which effectively suppressed the formation of intermetallic compounds. This effectively improved plasticity. For the present (TiVCrZr)_100−*x*_W*_x_* RHEAs, due to the larger negative mixing enthalpy between Zr and other compositional elements, it was more likely to result in the formation of the Zr-containing Laves phase. Therefore, reducing Zr content could be effective to suppress the formation of the Laves phase, further improving the plasticity of the (TiVCrZr)_100−*x*_W*_x_* RHEAs.

## 5. Conclusions

In this work, (TiVCrZr)_100−*x*_W*_x_ (x* = 5, 10, 15 and 20) RHEAs with a high specific strength were developed and investigated. The phase formation, microstructure and mechanical properties were examined. The main findings are summarized as follows:(1)(TiVCrZr)_100−*x*_W*_x_* RHEAs have the Laves, BCC and Ti-rich phases, where the Laves phase is the major phase, and the volume fraction of the BCC phase increases with increasing W content. Dendrite structures were observed. The EDS results show that W is enriched, and then gradually diffuses, and finally precipitates in the dendrite region, corresponding to the transformation from the gray region (Zr(Cr, V)_2_-type Laves phase) to bright region (Zr(Cr, V, W)_2_-type Laves phase) and the increased white region.(2)Empirical parameters and the calculated phase diagram (Thermo-Calc Software AB) are beneficial for understanding the mechanisms of phase formation and variation. The results indicate that the calculated phase type and variation are consistent with the experimental results. Moreover, the calculated phase diagram confirmed the substitution of W for Cr and V to some extent.(3)W addition significantly enhances the strength and hardness of (TiVCrZr)_100−*x*_W*_x_* RHEAs. The strength of the alloy system improved from 1574 MPa (W5) to 1894 MPa (W15), and the hardness improved from 588 HV to 694 HV. Generally, the higher specific strength benefits those alloys to be envisaged as structure materials. Compared with the selected RHEAs and Inconel 718, the specific strength of (TiVCrZr)_100−*x*_W*_x_*RHEAs exceed that of almost all W-containing RHEAs, and the highest specific strength was found to be 256.65 MPa·cm^3^·g^−1^ of W10 RHEA, which was just below that of the Mo_0.5_VNbTiCr_1.5_ and Mo_0.5_VNbTiCr_2.0_ W-free RHEAs. Moreover, the specific strength of (TiVCrZr)_100−*x*_W*_x_*RHEAs also exceed that of Inconel 718 (about 148 MPa·cm^3^·g^−1^), which is widely applied in the aerospace industry. The improvement of strength and hardness resulted from the strengthening effect of W on the Laves phase.

## Figures and Tables

**Figure 1 entropy-25-00100-f001:**
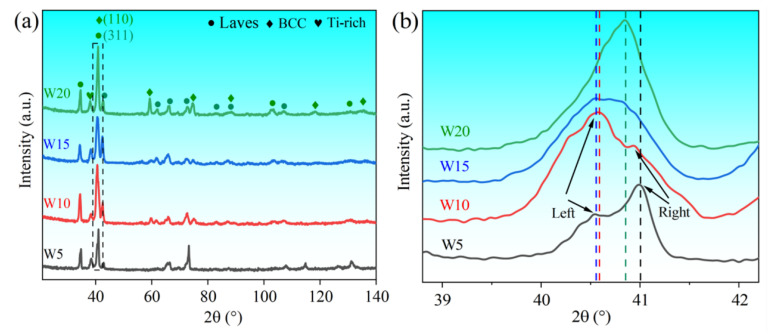
(**a**) XRD patterns of (TiVCrZr)_100−*x*_W*_x_* RHEAs; (**b**) the magnified XRD patterns near peak (311) marked in (**a**) (B-spline curve fitting).

**Figure 2 entropy-25-00100-f002:**
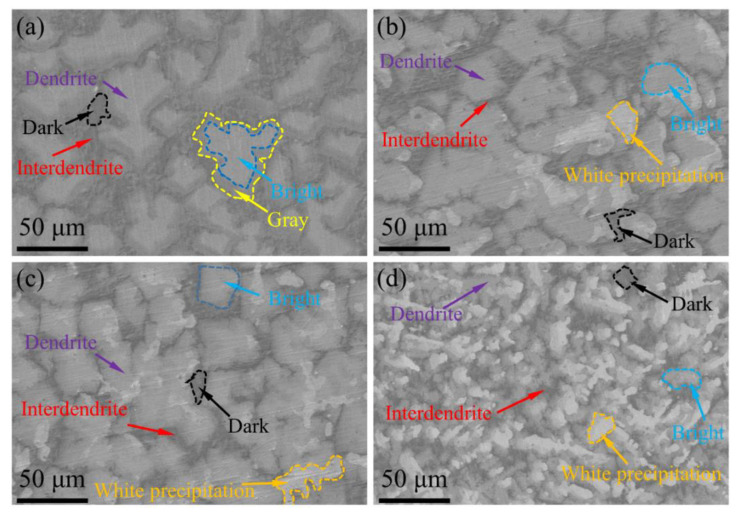
SEM images showing the microstructure of (TiVCrZr)_100−*x*_W*_x_* RHEAs: (**a**) W5; (**b**) W10; (**c**) W15 and (**d**) W20 RHEAs.

**Figure 3 entropy-25-00100-f003:**
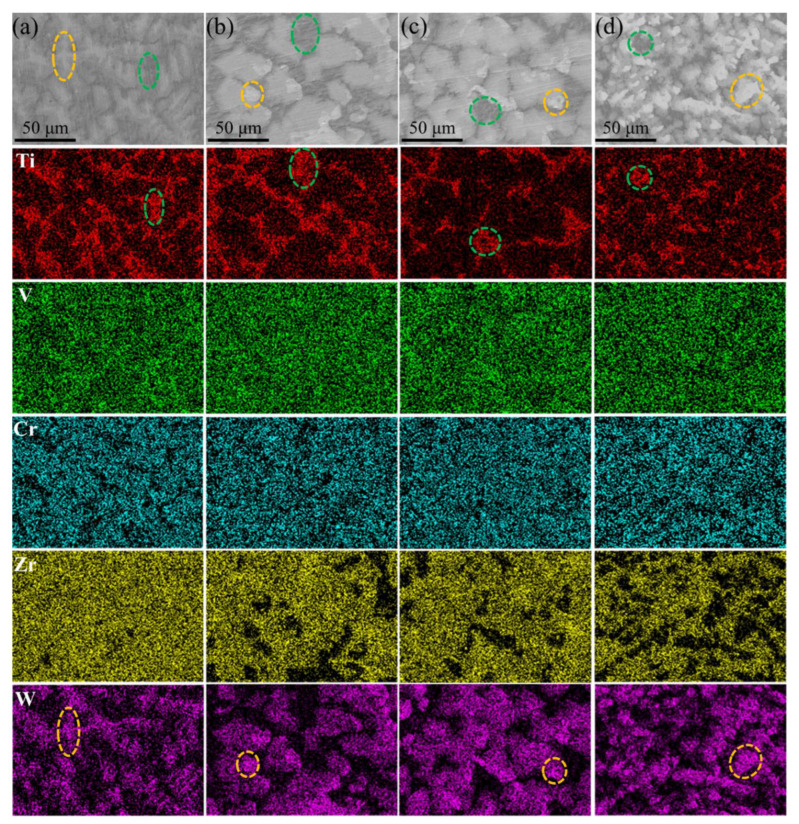
The EDS mapping results of (TiVCrZr)_100−*x*_W*_x_* RHEAs: (**a**) W5; (**b**) W10; (**c**) W15 and (**d**) W20 RHEAs.

**Figure 4 entropy-25-00100-f004:**
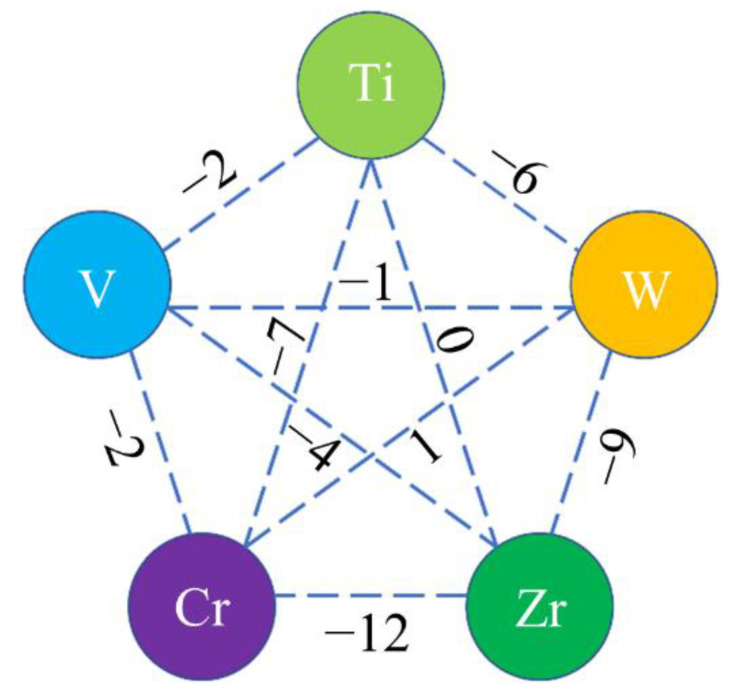
Diagram showing the enthalpy of mixing (kJ·mol^−1^) between pairs of atoms in (TiVCrZr)_100−*x*_W*_x_* RHEAs.

**Figure 5 entropy-25-00100-f005:**
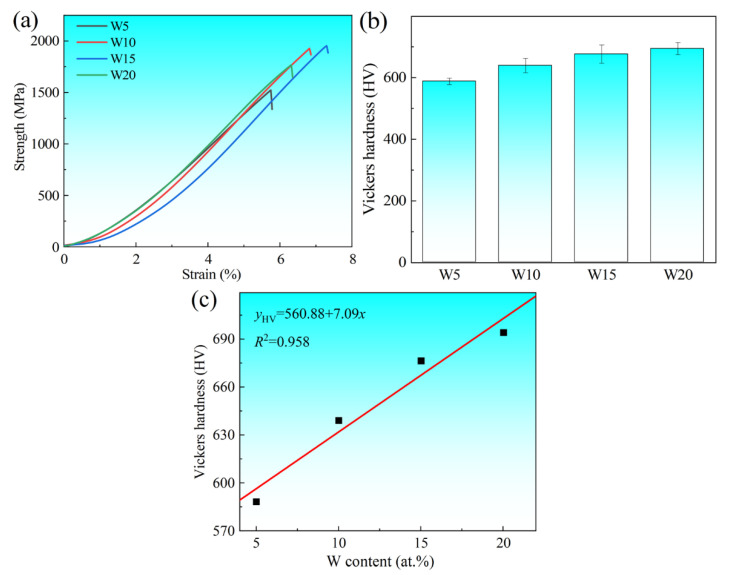
(**a**) Compressive stress-strain curves of (TiVCrZr)_100−*x*_W*_x_* RHEAs; (**b**) Vickers hardness of (TiVCrZr)_100−*x*_W*_x_* RHEAs and (**c**) the fitting curve of Vickers hardness and W content (at.%).

**Figure 6 entropy-25-00100-f006:**
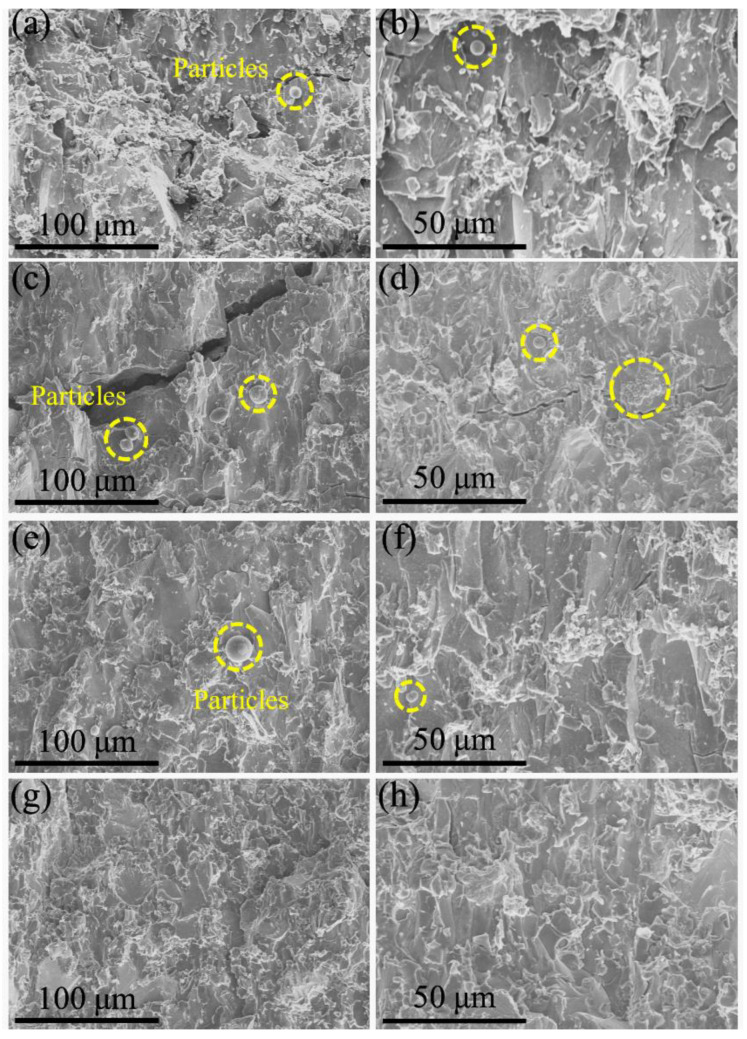
SEM images showing the fracture morphology of (TiVCrZr)_100−*x*_W*_x_* RHEAs: (**a**,**b**) W5; (**c**,**d**) W10; (**e**,**f**) W15 and (**g**,**h**) W20 RHEAs.

**Figure 7 entropy-25-00100-f007:**
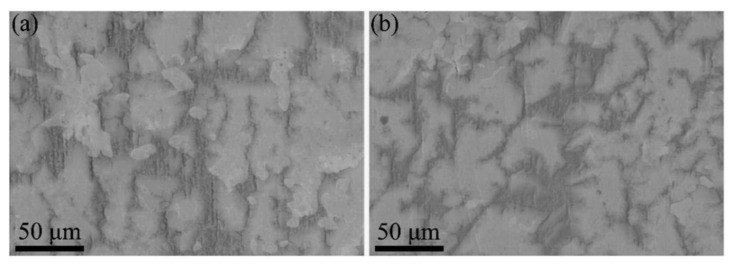
SEM images showing the microstructure of annealed and as-prepared W10 RHEA: (**a**) annealed; (**b**) as-prepared.

**Figure 8 entropy-25-00100-f008:**
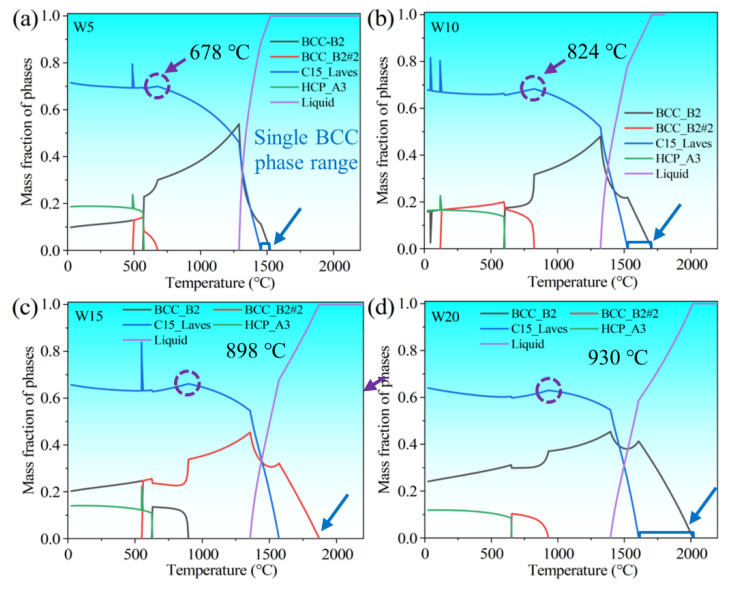
The calculated phase diagram showing the equilibrium solidification process of (TiVCrZr)_100−*x*_W*_x_* RHEAs: (**a**) W5; (**b**) W10; (**c**) W15 and (**d**) W20 RHEAs.

**Figure 9 entropy-25-00100-f009:**
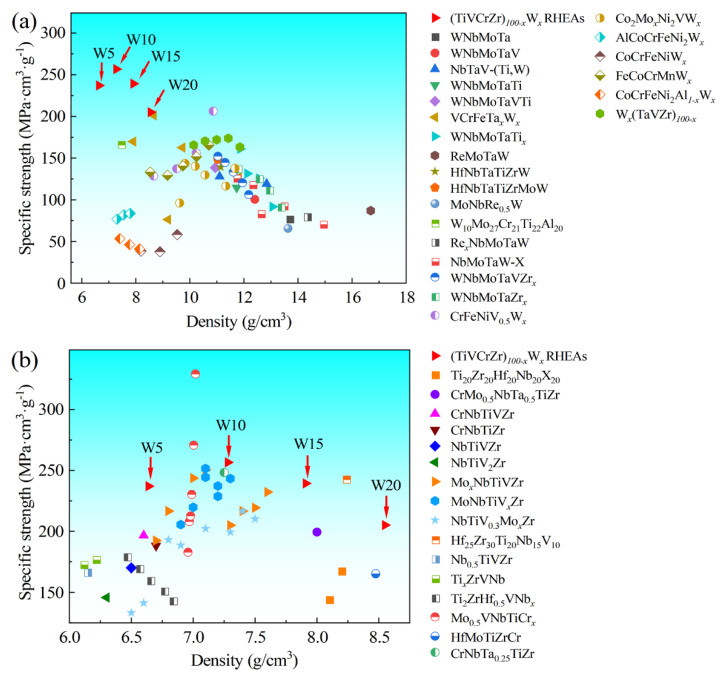
Plot of comparison of the specific strength and density of (TiVCrZr)_100−*x*_W*_x_* RHEAs and partial RHEAs: (**a**) (TiVCrZr)_100−*x*_W*_x_* RHEAs and W-containing RHEAs [9,10,15,26,27,28,40,57,58,59,60,61,62,63,64,65,66,67,68,69]; (**b**) (TiVCrZr)_100−*x*_W*_x_* RHEAs and W-free RHEAs [8,11,24,38,70,71,72,73,74,75,76,77].

**Table 1 entropy-25-00100-t001:** The elemental contents of the distinct regions of (TiVCrZr)_100−*x*_W*_x_* RHEAs.

RHEAs	Region	Content (at.%)				
		Ti	V	Cr	Zr	W
	Bright	9.54 ± 0.46	19.92 ± 0.97	26.36 ± 0.86	29.66 ± 0.47	14.52 ± 1.27
W5	Gray	14.60 ± 0.80	24.78 ± 0.44	30.04 ± 0.59	26.82 ± 0.18	3.82 ± 0.35
	Dark	55.20 ± 1.56	18.55 ± 1.06	5.25 ± 0.78	20.6 ± 0.42	0.45 ± 0.07
	White	14.08 ± 0.15	23.78 ± 0.46	10.50 ± 0.70	0.8 ± 0.47	50.85 ± 1.06
W10	Bright	10.13 ± 0.57	19.4 ± 0.83	24.6 ± 0.57	29.9 ± 0.29	16.00 ± 0.43
	Gray	17.1 ± 17.1	24.3 ± 0.85	29.5 ± 0.99	27.3 ± 0.85	1.8 ± 0.28
	Dark	55.20 ± 0.80	15.23 ± 0.68	4.93 ± 0.55	23.87 ± 2.10	0.67 ± 0.15
	White	12.93 ± 0.68	21.37 ± 0.42	10.2 ± 0.36	1.77 ± 0.74	53.73 ± 0.90
W15	Bright	7.36 + 0.47	17.03 + 0.68	23.57 + 0.51	30.9 + 0.36	21.17 + 1.49
	Dark	54.92 ± 0.67	15.08 ± 0.87	5.33 ± 0.49	24.05 ± 1.06	0.6 ± 0.14
	White	11.03 ± 0.57	15.23 ± 0.96	7.13 ± 0.51	1.43 ± 0.55	65.2 ± 0.75
W20	Bright	9.65 + 1.77	18.05 + 1.20	24.25 + 0.35	30.2 + 0.14	17.75 + 2.47
	Dark	55.68 ± 1.50	14.84 ± 1.91	5.38 ± 0.92	23.00 ± 3.76	1.1 ± 0.42

**Table 2 entropy-25-00100-t002:** The fracture strength (*σ*_b_), strain (*ε*_f_) and Vickers hardness (HV) of (TiVCrZr)_100−*x*_W*_x_* RHEAs at room temperature.

Alloy	*σ*_b_ (MPa)	*ε*_f_ (%)	HV (kgf/mm^2^)
W5	1574 ± 56.9	5.94 ± 0.5	588 ± 10
W10	1867 ± 52.5	6.68 ± 0.9	639 ± 23
W15	1894 ± 51.6	7.42 ± 0.6	676 ± 29
W20	1754 ± 34.2	6.47 ± 0.6	694 ± 20

**Table 3 entropy-25-00100-t003:** The comparison of hardness and strength of the annealed and as-prepared W10 RHEA.

Alloy	*σ*_b_ (MPa)	HV (kgf/mm^2^)
As-prepared (W10)	1867 ± 52.5	639 ± 23
Annealed (W10)	1930 ± 45	647 ± 13

**Table 4 entropy-25-00100-t004:** Several parameter values of the elements of (TiVCrZr)_100−*x*_W*_x_* RHEAs [45].

Element	Ti	V	Cr	Zr	W
*r* (Å)	1.462	1.316	1.249	1.603	1.367
*χ* _Allen_	1.38	1.53	1.65	1.32	1.47
*VEC*	4	5	6	4	6
*a* (Å)	3.31	3.03	2.91	3.58	3.17
*T*_m_ (K)	1941	2183	2180	2128	3695
*ρ* (g/cm^3^)	4.51	6.11	7.14	6.51	19.25

**Table 5 entropy-25-00100-t005:** The calculated empirical parameter values of (TiVCrZr)_100−*x*_W*_x_* RHEAs.

Alloy	Δ*S*_mix_ [J/(K·mol)]	Δ*H*_mix_ (kJ/mol)	*δ* (%)	*VEC*	*Ω*	*T*_m_ (K)	Δ*χ*_Allen_ (%)
W5	12.600	−6.804	9.497	4.813	4.05	2187.4	8.556
W10	13.076	−6.817	9.277	4.875	4.35	2266.7	8.327
W15	13.312	−6.789	9.048	4.938	4.60	2346.1	8.093
W20	13.382	−6.720	8.810	5.000	4.83	2425.4	7.851

**Table 6 entropy-25-00100-t006:** The specific strength and *ρ* of (TiVCrZr)_100−*x*_W*_x_* RHEAs.

Alloys	W5	W10	W15	W20
The specific strength (MPa·cm^3^·g^−1^)	237.03	256.65	239.29	205.10
*ρ* (g/cm^3^)	6.64	7.28	7.91	8.55

## Data Availability

The raw/processed data will be made available on request.

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
