# Peer review of "Microstructure and Mechanical Properties of High-Specific-Strength (TiVCrZr)100−xWx (x = 5, 10, 15 and 20) Refractory High-Entropy Alloys"

_entropy, 2023, doi:10.3390/e25010100_

Round 1
Reviewer 1 Report
The authors investigate the microstructure (XRD, SEM+EDX), and mechanical properties (HV and compression) of (TiVCrZr)_(100-x)W_x (x = 5, 10, 15 and 20) high entropy alloys. The field of HEA compositions is very large (and also not limited to metallic materials) and thus any investigations performed on HEAs are prone to develop the knowledge base for them. The present work also add some points to this base, but as a scientific paper it has important flaws and should be consistently improved to be accepted for publication.
Some of the weak points are listed below:
1) The possible applications of these materials are very general and therefore also the investigations performed lack of relevance and confusing. E.g. mechanical properties for refractory materials investigated only at room temperature.
2) The investigated materials should be also compared in the conclusions' section with other state of the art materials, not only among them. At least based on Fig. 8 points.
3) There are also punctual statements which should be corrected/improved or better explained.
a) line 87, grits 400-5000: which kind of grits (US, FEPA, ...) ? should be mentioned. Same on line 93.
b) line 90, please also mention the producer of HV tester
c) line 95, what testing machine (model, producer)
d) line 113 and fig 1 (b): a fitting will be more convincing
e) lines 148-150, the statement is speculative, and sure not general, since both melting process and cooling speeds are very important for this.
f) lines 179-180 and 182-184, see point d) above
g) line 207, the linear fit is clearly a poor approximation. Please reconsider.
h) line 231 and fig 6, does not look to me as a typical brittle fracture morphology, maybe better contrast images might help.
i) line 293-294: Please elaborate more on the phase diagram calculation, and mention which program was used
j) line 338 and eq. 9: although "almost consistent" the theoretical density should also be checked by using XRD data, there are several phases observed at RT, so the mixing rule should be applied to the phase mixture.
h) line 351-352. "... while obtaining low density'" Low compared to what material? 8.55 is pretty high, is it not?
k) line 395 "designed" is not justified by the paper text, should be replaced by a more appropiate term.
l) line 403 please mention the calculation program for the phase diagram
With these modifications I expect the paper should be reconsidered for publication.
Reviewer 2 Report
I humbly note that it is impolite to list Cantor's paper as first and Jien-Wei Yeh's as sixth, as if relegating Yeh's priority to the discovery of high-entropy alloys. As I recall, the goal of the Cantor group was to prove the role of confusion principle (Lindsay Greer) in helping the bulk amorphous forming ability. Confusion was created by increasing the number of constituents up to 20. On the other hand, Jien-wei Yeh increased the number of constituents in order to stabilize the Gibbs free energy with the contribution of the configurationally entropy term and thereby promote the creation of the extended solid solution. Moreover, Yeh patented the idea in 2002 and the article was published only in 2004, the same year as the Cantor article. Both arrived to the multcomponent alloys but on quite different way of thinking.
A major disadvantage of ETM-based refractory alloys is that they cannot withstand oxidation. There is no mention of this in the article. Another barrier to applications is fragility. Please comment on the oxidation resistance and how to improve the ductility of these alloys.
Author Response
Responses to Reviewer #2
Comment 1. I humbly note that it is impolite to list Cantor's paper as first and Jien-Wei Yeh's as sixth, as if relegating Yeh's priority to the discovery of high-entropy alloys. As I recall, the goal of the Cantor group was to prove the role of confusion principle (Lindsay Greer) in helping the bulk amorphous forming ability. Confusion was created by increasing the number of constituents up to 20. On the other hand, Jien-wei Yeh increased the number of constituents in order to stabilize the Gibbs free energy with the contribution of the configurationally entropy term and thereby to promote the creation of the extended solid solution. Moreover, Yeh patented the idea in 2002 and the article was published only in 2004, the same year as the Cantor article. Both arrived to the multcomponent alloys but on quite different way of thinking.
Responses: Thank you for the good suggestion. Your suggestion helped us better understand the development of HEAs, which will be extremely valuable for our later research on HEAs. The order of the references has been revised.
Comment 2. A major disadvantage of ETM-based refractory alloys is that they cannot withstand oxidation. There is no mention of this in the article. Another barrier to applications is fragility. Please comment on the oxidation resistance and how to improve the ductility of these alloys.
Responses: Thank you for the good suggestion.
On the one hand, the oxidation resistance of materials is indeed a very important performance indication. A recent work has summarized the functional application potential of W-containing RHEAs (Ref. [5] in this paper). It indicated that the addition of Ti and Cr elements can significantly improve the oxidation resistance of materials, suggesting that (TiVCrZr)100-xWx RHEAs may have better oxidation resistance. Thus, the present work indicated that (TiVCrZr)100-xWx RHEAs could also have the potential for high-temperature applications. Moreover, in the revised paper, annealing of the typical W10 RHEAs has been conducted, where the annealed specimen demonstrated similar microstructure and slightly increased strength and hardness. This further validates the possibility for high-temperature applications. However, the main purpose of this work was to develop a RHEA system with a relatively high specific strength. The oxidation resistance of investigated materials may be further studied in next stage. In the revised paper, following contents were made:
- in the original manuscript, Ref. [4] has been replaced by the recent review article (Chen, S.H., Qi, C., Liu, J.Q. et al. Recent Advances in W-Containing Refractory High-Entropy Alloys—An Overview. Entropy.).
- Annealing of the typical W10 RHEA was conducted. And the detail of the changes is shown in Response to Comment 1 of Reviewer #
On the other hand, improving the plasticity of (TiVCrZr)100-xWx RHEAs is also of great significance. In this work, the purpose of getting high specific strength has been achieved. Improving the plasticity is challenging due to the strength-ductility trade-off. However, in literature, many studies have proposed some methods to further improve the plasticity by maintaining the strength (Ref. [28,56,58] in this paper), for example, Sun et al. tuned elemental composition to minimize the Gibbs free energy, which effectively suppressed the formation of intermetallic compounds. This effectively improves plasticity. For the present (TiVCrZr)100-xWx RHEAs, due to the larger negative mixing enthalpy between Zr and other compositional elements, as shown in Fig. 4 in the original manuscript, it was more likely to result in the formation of the Zr-containing Laves phase. Therefore, reducing Zr content may be effective to suppress the formation of the Laves phase, further improving plasticity. In the revised paper, following discussion were added at the section Discussion:
“Improving the plasticity of the (TiVCrZr)100-xWx RHEAs could expand their engineering applications. However, improving the plasticity is challenging due to the strength-ductility trade-off. Many ways have been proposed to further improve the plasticity by maintaining the strength [28,56,58], for example, Sun et al. [56] tuned elemental composition to minimize the Gibbs free energy, which effectively suppressed the formation of intermetallic compounds. This effectively improved plasticity. For the present (TiVCrZr)100-xWx RHEAs, due to the larger negative mixing enthalpy between Zr and other compositional elements, it was more likely to result in the formation of the Zr-containing Laves phase. Therefore, reducing Zr content could be effective to suppress the formation of the Laves phase, further improving the plasticity of the (TiVCrZr)100-xWx RHEAs.”
Round 2
Reviewer 2 Report
Concerning the oxidation resistance and plasticity I have got texts instead of facts